# Context-dependent relationships between locus coeruleus firing patterns and coordinated neural activity in the anterior cingulate cortex

**Siddhartha Joshi\*, Joshua I Gold**

Department of Neuroscience, University of Pennsylvania, Philadelphia, United States

**Abstract** Ascending neuromodulatory projections from the locus coeruleus (LC) affect cortical neural networks via the release of norepinephrine (NE). However, the exact nature of these neuro-modulatory effects on neural activity patterns in vivo is not well understood. Here, we show that in awake monkeys, LC activation is associated with changes in coordinated activity patterns in the anterior cingulate cortex (ACC). These relationships, which are largely independent of changes in firing rates of individual ACC neurons, depend on the type of LC activation: ACC pairwise correlations tend to be reduced when ongoing (baseline) LC activity increases but enhanced when external events evoke transient LC responses. Both relationships covary with pupil changes that reflect LC activation and arousal. These results suggest that modulations of information processing that reflect changes in coordinated activity patterns in cortical networks can result partly from ongoing, context-dependent, arousal-related changes in activation of the LC-NE system.

## Editor's evaluation

This is a timely and important study that systematically assesses the relationships between neuronal activity in the locus coeruleus (LC) and the anterior cingulate cortex (ACC) in non-human primates. The LC is a major source of cortical norepinephrine that has reciprocal connectivity with the ACC, and the authors have convincingly shown that LC spiking is associated with changes in ACC spike correlations. Further, these changes have consistent phase relationships with pupil size. This is a rare data set that is technically challenging to acquire, and the results are an important advance toward understanding a circuit that is likely to play a role in regulating brain states such as arousal or attention.

**\*For correspondence:**
thesidjoshi@gmail.com

## Introduction

Changes in brain state are associated with different levels of arousal, attention, motivation, surprise, and other factors that can affect the activity patterns of large populations of cortical neurons (*McAdams and Maunsell, 1999*; *Purcell et al., 2012*; *Chang et al., 2012*; *Falkner et al., 2013*; *Downer et al., 2015*; *Ecker et al., 2016*; *Thiele et al., 2016*). These changes are thought to result, in part, from the widespread release of neuromodulators (*Aoki et al., 1987*; *Devauges and Sara, 1990*; *Schultz et al., 1993*; *Dalley et al., 2001*; *Bouret and Sara, 2005*; *Aston-Jones and Cohen, 2005*; *McGaughy et al., 2008*; *Salamone et al., 2009*; *Pinto et al., 2013*; *Varazzani et al., 2015*; *Khani and Rainer, 2016*; *Minces et al., 2017*). Different neuromodulatory systems have different anatomical and physiological properties (*Kupfermann, 1979*; *Aoki et al., 1987*; *Xiang et al., 1998*; *Flores-Hernandez et al., 2000*; *Fernández-Pastor and Meana, 2002*; *Ding and Perkel, 2002*; *Salgado et al., 2011*; *Herrero et al.,*

*2013*; *Sugihara et al., 2016*; *Doyle and Meeks, 2017*) that are thought to support their different roles in neural information processing (*Abbott and Dayan, 1999*; *Nirenberg and Latham, 2003*; *Averbeck et al., 2006*; *Silver, 2010*; *Beck et al., 2011*; *Moreno-Bote et al., 2014*; *Kanitscheider et al., 2015*; *Lin et al., 2015*; *Rodenkirch et al., 2019*). One prominent example is the locus coeruleus (LC)-norepinephrine (NE) system, whose diffuse projections throughout the brain, close relationship to the sleep-wake cycle, and relationship to electroencephalography (EEG) and pupillometry have led to several theories of its role in arousal-related modulations of cortical activity and function (*Aston-Jones and Cohen, 2005*; *Aston-Jones and Bloom, 1981*; *Nieuwenhuis et al., 2011*; *Gilzenrat et al., 2010*; *Einhäuser et al., 2010*; *Murphy et al., 2011*; *Murphy et al., 2014*; *Joshi et al., 2016*; *Joshi and Gold, 2020*). However, only a small number of studies have shown direct relationships between (artificial) LC activity and cortical activity in vivo (e.g., *Devilbiss and Waterhouse, 2011*), limiting our understanding of the exact nature of these relationships.

Our aim was to test if and how endogenous, ongoing activity and sensory-driven, evoked responses in the LC relate to changes in neural activity patterns in the anterior cingulate cortex (ACC) of the primate brain. We targeted the ACC because it has strong reciprocal connectivity, both structural and functional, with the LC (*Lewis et al., 1979*; *Morrison et al., 1979*; *Porrino and Goldman-Rakic, 1982*; *Jones and Olpe, 1984*; *Fernández-Pastor et al., 2005*; *Gompf et al., 2010*; *Chandler and Water-house, 2012*; *Chandler et al., 2013*; *Tervo et al., 2014*; *Köhler et al., 2016*; *De Gee et al., 2017*; *Koga et al., 2020*). Moreover, ACC neural activity can encode computations that underlie adaptive, goal-directed behaviors, including those that are also associated with indirect measures of LC-linked arousal, such as changes in pupil size and the P300 component of the event-related potential (ERP; *Shima and Tanji, 1998*; *Gehring and Willoughby, 2002*; *Holroyd and Coles, 2002*; *Critchley et al.,*

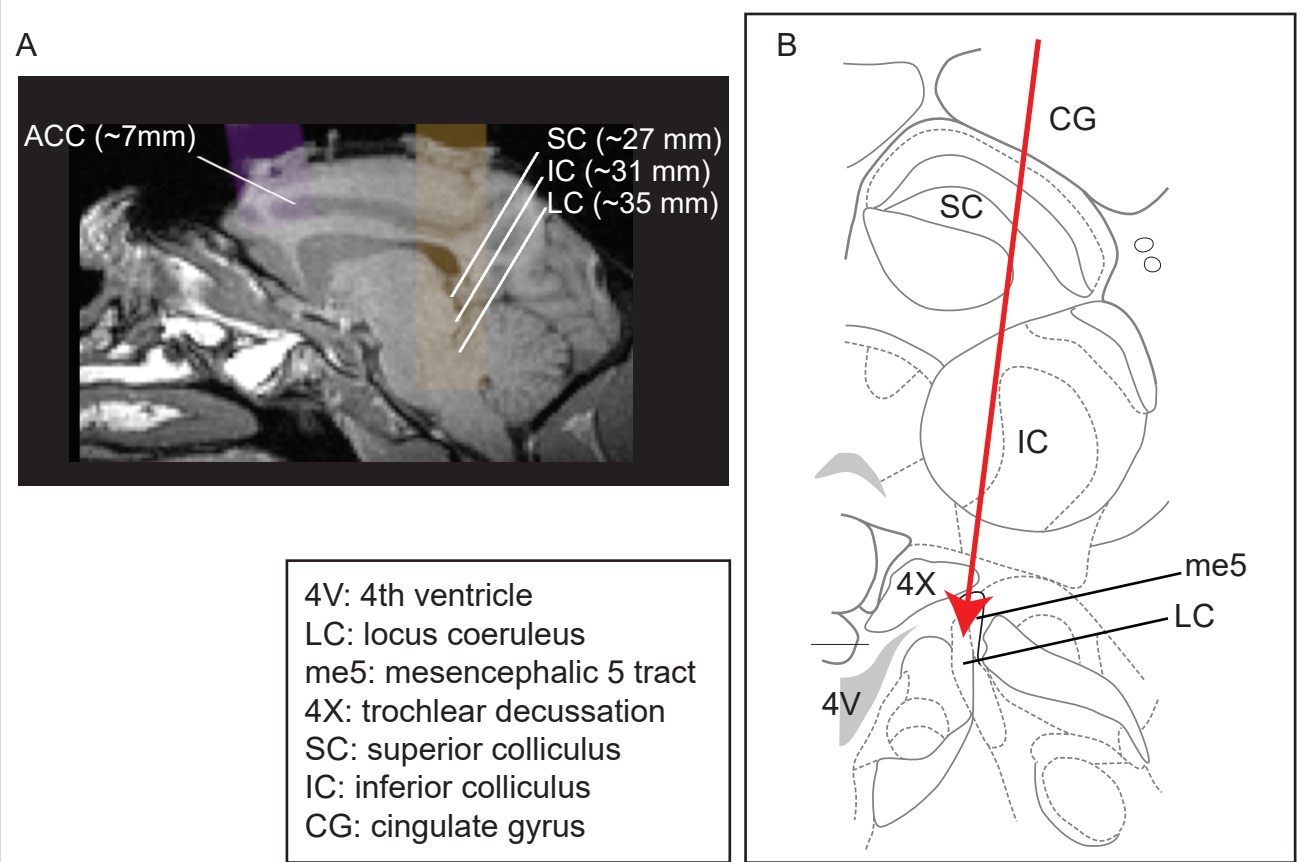

**Figure 1.** Recording site locations. (**A**) Approximately sagittal MRI section showing targeted recording locations in the anterior cingulate cortex (ACC) (areas 32, 24b, and 24c) and locus coeruleus (LC) for monkey Ci (right), with the SC and IC shown for reference. For recording locations in monkeys, Oz, Sp, and Ci (left hemisphere), see *Kalwani et al., 2014*; *Joshi et al., 2016*. (**B**) Schematic of a coronal section showing structures typically encountered along electrode tracts to LC (adapted from *Paxinos et al., 2008*; Plate 90, Interaural 0.3; bregma 21.60).

*2005*; *Matsumoto et al., 2007*; *Hayden et al., 2011*; *Shenhav et al., 2013*; *Ebitz and Platt, 2015*; *Sarafyazd and Jazayeri, 2019*). Thus, interactions between LC and ACC neural activity patterns are likely to have broad behavioral relevance.

We recorded neural activity simultaneously in LC and ACC (*Figure 1*) and measured the pupil size of alert monkeys under two behavioral conditions: (1) performing a fixation task and (2) performing a fixation task with randomly presented sounds. Both of these conditions have been associated with variations in arousal that covary with pupil size and can affect cognition and behavior (*Aston-Jones and Bloom, 1981*; *Nieuwenhuis et al., 2011*; *Einhäuser et al., 2010*; *Gilzenrat et al., 2010*; *Murphy et al., 2011*; *Murphy et al., 2014*; *Varazzani et al., 2015*; *Joshi et al., 2016*). Conditioned on the firing (or lack of firing) of LC neurons, we measured activity patterns of individual neurons and coordinated activity between pairs of neurons in ACC, both of which govern the information processing capacities of neural networks (*Britten et al., 1992*; *Parker and Newsome, 1998*; *Cohen and Kohn, 2011*; *Moreno-Bote et al., 2014*; *Lin et al., 2015*; *Kohn et al., 2016*). We were particularly interested in the timescales over which these features of neural activity in LC and ACC are related, which can provide insights into putative underlying mechanisms. For example, neuromodulatory effects of the LC-NE system on ACC might be expected to have a relatively long time course compared with the time course of typical synaptic events that are mediated by faster, glutamatergic neurotransmission (*Feldman RS et al., 1997*; *McCormick and Prince, 1988*; *Wang and McCormick, 1993*; *Schmidt et al., 2013*; *Timmons et al., 2004*). Below we show that such relatively long timescale relationships are evident between LC and ACC neural activity patterns, in particular changes in coordinated activity between pairs of ACC neurons that tend to decrease or increase in relation to spontaneous or evoked LC spikes, respectively.

## Results

We analyzed neural activity from simultaneously measured sites in the LC (n = 84 single units, including 78 simultaneously recorded pairs from 35 sites in monkey Sp; 84/80/31 in monkey Ci) and ACC (372/4336/35 in monkey Sp; 275/2875/31 in monkey Ci) and from LC-only recordings (71/12/73 in monkey Ci and 36/8/34 in monkey Oz) while the monkeys maintained fixation on a visual spot. Our analyses focused on whether and how mean spiking activity and individual and pairwise neuronal variability measured in one brain region (ACC or LC) related to neuronal activity in the other brain region (LC or ACC, respectively). To understand the temporal dynamics of these potential relationships, we systematically tested for effects across a broad range of time windows. These windows ranged in duration from 100 ms, which is often used to study neural synchrony, to 1 s, which is consistent with typical timescales of slow neuromodulatory influences (*Feldman RS et al., 1997*; *McCormick and Prince, 1988*; *Wang and McCormick, 1993*; *Schmidt et al., 2013*; *Timmons et al., 2004*; *Doiron et al., 2016*). Because LC and ACC are connected reciprocally, we tested for relationships in both directions. Reliable relationships in either direction are included in the main figures, and the remaining results are shown in figure supplements.

### Single-neuron activity during passive fixation

During passive fixation, LC neurons were weakly active, with no spikes measured on more than a third of all trials and otherwise a median (interquartile range [IQR]) firing rate of 1.8 [0.9–3.6] sp/s measured during 1.1 s of stable fixation (*Figure 2A*). For each session, we divided trials into six groups based on the given LC neuron's firing rate: trials with firing rate = 0 (LC$_{zero}$), 1, 2, 3, and ≥4 sp/s formed five groups, and all trials with firing rate >0 formed a sixth group (LC$_{non-zero}$). We then assessed ACC neural activity measured at the same time, conditioned on the LC group.

The mean and variance of ACC spike counts, along with their Fano factor (the ratio of variance over mean), increased steadily as a function of the duration of the counting window. These trends showed an apparent, small relationship with simultaneously measured LC spiking activity, such that the mean, variance, and Fano factor measured in the ACC were, on average, slightly higher for LC$_{zero}$ versus LC$_{non-zero}$ trials (*Figure 2B–D*). However, these relationships were not statistically reliable when considering data from both monkeys combined together and only in one case (*Figure 2G*) when considering data separately for each monkey (sign-rank test for $H_0$: median difference between LC$_{zero}$ and LC$_{non-zero}$ conditions computed per ACC unit, or ANOVA test for an effect of LC firing rate group, p > 0.05 in all

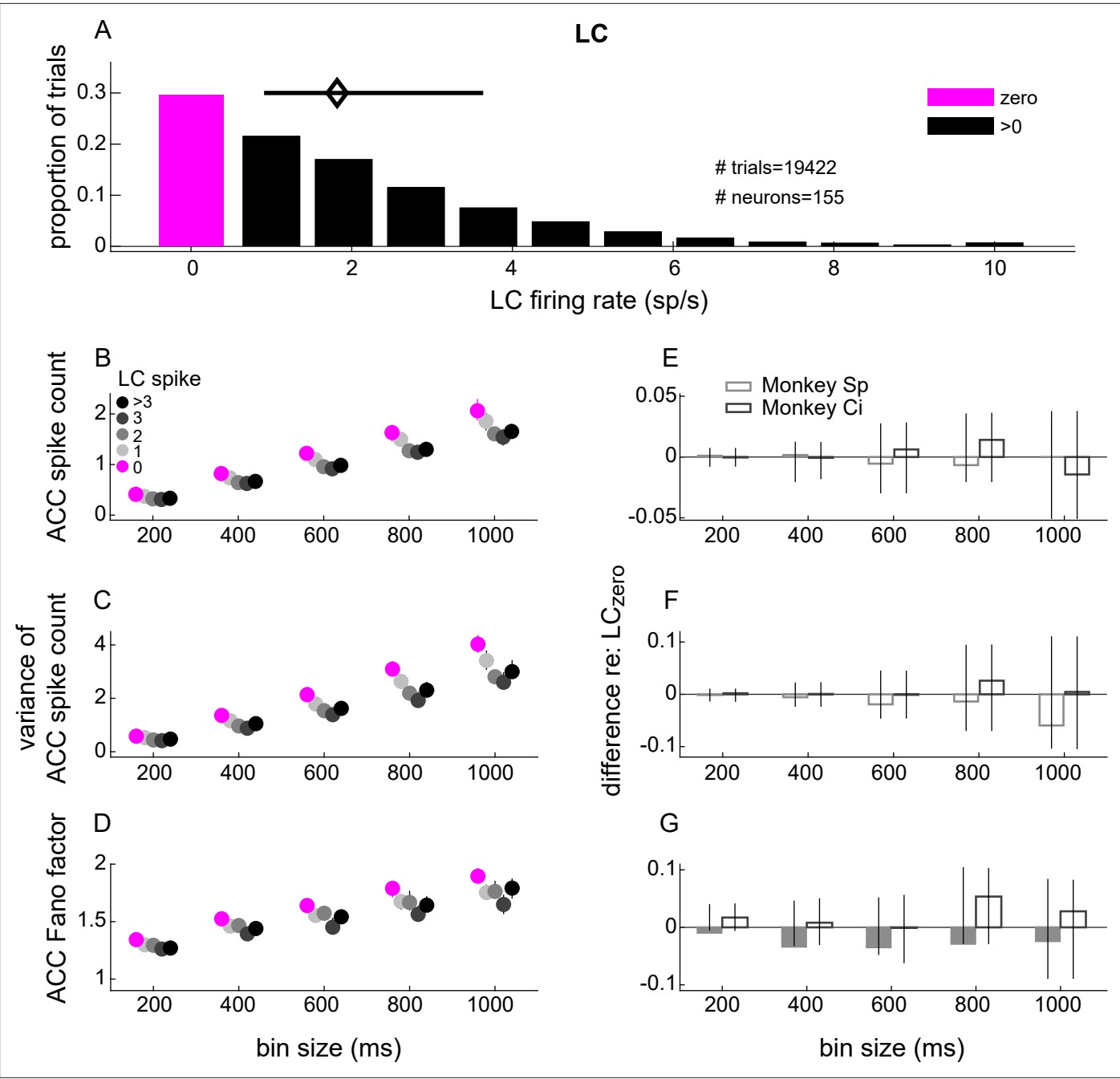

**Figure 2.** Anterior cingulate cortex (ACC) single-unit spike-count statistics conditioned on simultaneously measured locus coeruleus (LC) spiking. (**A**) LC single-unit firing rate distribution measured in 1.1 s windows starting 1 s after the onset of stable fixation from all trials and recording sessions. The magenta and black bars indicate the proportion of trials with 0 and >0 LC spikes, respectively. The diamond and horizontal bar indicate median and interquartile range (IQR), respectively, from trials with ≥1 spike. (**B–D**) ACC single-unit spike count (**B**), variance of the spike count (**C**), and Fano factor (variance/mean; **D**) from trials in which the simultaneously measured LC unit spiked as indicated in the legend in (**B**). ACC spikes were counted in five equally spaced bins ranging from 200 ms to 1 s. Symbols and error bars are the median and bootstrapped 95% confidence interval of the distribution of values computed per ACC unit, pooling the data across all units recorded from both monkeys. (**E–G**) Difference between each value from panels (**A–C**), respectively, measured between the LC > 0 condition and the LC = 0 condition. Bars and error bars are the median and bootstrapped 95% confidence interval of the distribution of values computed per ACC unit. Filled bars and symbols above indicate $p < 0.05$ for sign-rank tests for $H_0$: median difference between $LC_{zero}$ and $LC_{non-zero}$ conditions = 0 tested: (1) separately for each monkey (filled bars) and (2) using data combined from both monkeys (*none found).

The online version of this article includes the following figure supplement(s) for figure 2:

*Figure 2 continued on next page*

*Figure 2 continued*

**Figure supplement 1.** Anterior cingulate cortex (ACC) single-neuron spike count and variability conditioned on locus coeruleus (LC) spiking (measured as differences between measurements from $LC_{non-zero}$ versus $LC_{zero}$ trials).

**Figure supplement 2.** Locus coeruleus (LC) single-unit spike-count statistics conditioned on simultaneously measured anterior cingulate cortex (ACC) spiking.

cases; *Figure 2B–G*). Across ACC units, differences in Fano factor for $LC_{zero}$ versus $LC_{non-zero}$ trials were associated strongly with differences in spike-count variance and more weakly with differences in spike counts (*Figure 2—figure supplement 1*).

In addition, the mean, variance, and Fano factor of LC spike counts also increased steadily as a function of the duration of the counting window. However, none of these trends showed a statistically significant relationship to simultaneously measured ACC spiking activity (sign-rank test and ANOVA, p > 0.05 in all cases; *Figure 2—figure supplement 2*).

## Coordinated activity during passive fixation

During passive fixation, pairs of neurons in ACC had coordinated spiking activity that varied considerably in sign and magnitude across pairs but also, as has been reported previously, depended systematically on the size of the time window used to count spikes (*de la Rocha et al., 2007*). Specifically, pairwise spike-count correlations ($r_{sc}$) in ACC had median [IQR] values of 0.03 [–0.01 0.07] using 200 ms counting windows and systematically increased and became more variable across pairs with larger counting windows up to 1 s (0.05 [–0.03 0.14]; ANOVA test for effect of window size, p = 0.0243).

These ACC spike-count correlations also depended on the spiking activity of LC neurons measured at the same time. Specifically, the correlations were largest when (1) they were measured using relatively large time windows; (2) for ACC pairs that exhibited relatively large, positive correlations independent of LC firing; and (3) when the simultaneously recorded LC neuron was not active (*Figures 3 and 4*). For the example ACC pair illustrated in *Figure 3A–C*, $r_{sc}$ was near zero when computed using relatively small time windows and then increased steadily with increasing bin sizes. The magnitude of these increases in $r_{sc}$ with bin size was larger on trials in which the simultaneously measured LC neuron was not spiking versus spiking (*Figure 3C*).

The population of ACC pairs exhibited similar trends, with $r_{sc}$ values tending to be smaller on $LC_{non-zero}$ versus $LC_{zero}$ trials, particularly for larger time windows and for ACC pairs with non-negative LC-independent $r_{sc}$ values. To visualize these effects, we first divided ACC pairs into terciles, according to their LC-independent $r_{sc}$ values (*Figure 4*). Under these conditions, ACC $r_{sc}$ values from the upper two terciles of LC-independent $r_{sc}$ values were reduced by up to ~50% when computed on trials with at least one or more LC spike versus trials with no LC spikes (*Figure 4D–F*). The magnitude of these reductions was not related in a consistent manner to the magnitude of $LC_{non-zero}$ activity (ANOVA test for group effect of LC firing rates > 0). These relationships were not evident on shuffled trials, supporting the idea that the trial-by-trial relationships that we identified were not spurious reflections of the spiking statistics of each region considered separately (*Figure 4—figure supplement 1*). Furthermore, these LC-linked differences in ACC $r_{sc}$ values did not result simply from changes in ACC neuron firing rates because the two measures were unrelated (*Figure 4—figure supplement 2A*). We found a weak but statistically reliable relationship between LC-linked changes in ACC $r_{sc}$ and changes in ACC Fano factor (*Figure 4—figure supplement 2B*).

In contrast, spike-count correlations ($r_{sc}$) from simultaneously measured pairs of LC neurons did not depend systematically on concurrently measured ACC activity. In general, ACC-independent LC $r_{sc}$ values were roughly similar to LC-independent ACC $r_{sc}$ values, increasing steadily and becoming more variable as a function of the counting-window size; for example, median [IQR] values were 0.02 [0 0.09] for 200 ms windows and 0.06 [–0.02 0.19] for 1 s windows. However, these values did not show reliable differences when compared on trials with high versus low ACC firing, analyzed in the same way as the LC-linked ACC $r_{sc}$ values (*Figure 4—figure supplement 3*). Moreover, the distributions of these ACC-linked LC $r_{sc}$ values did not appear to come from the same (shifted) distribution as the LC-linked ACC $r_{sc}$ values for all five time bins (Kolmogorov–Smirnov test for $H_0$: both sets of values come from the same distribution, p < 0.0189 in all five cases). Thus, differences in single-unit LC activity were associated with differences in coordinated activity patterns of pairs of neurons in ACC, but differences

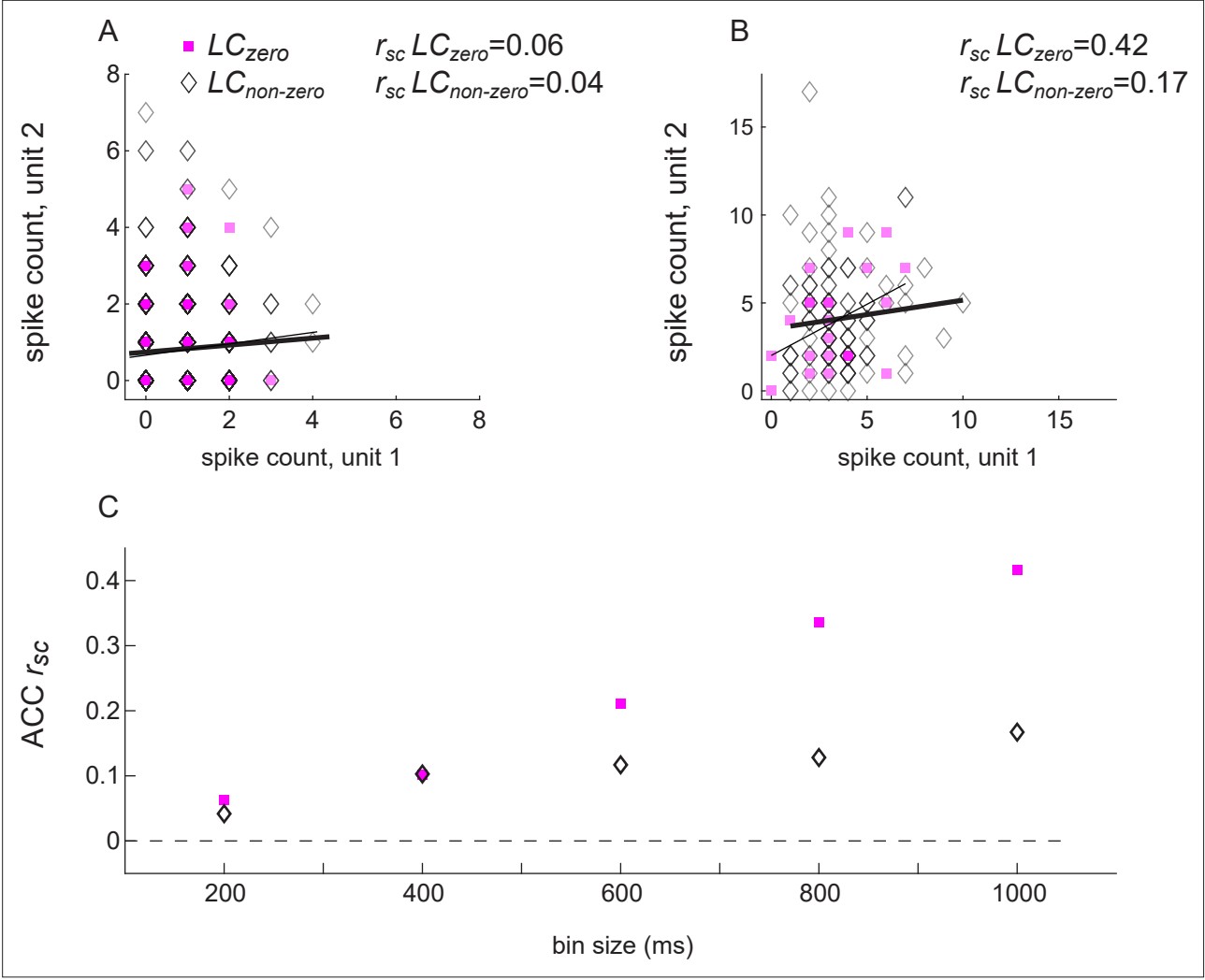

**Figure 3.** Spike-count correlations ($r_{sc}$) of an example anterior cingulate cortex (ACC) pair conditioned on the spiking activity of a simultaneously recorded locus coeruleus (LC) unit. (**A**) ACC spikes counted in a 200 ms-wide bin. (**B**) ACC spikes counted in a 1000 ms-wide bin. In (**A**) and (**B**), square/diamond markers indicate data from trials in which the simultaneously recorded LC unit had zero/non-zero firing rates. Thin/thick lines are linear fits to these data points, respectively. (**C**) Spike-count correlation ($r_{sc}$) for the example ACC pair shown in (**A**) and (**B**) as a function of bin size, computed separately for trials in which the simultaneously recorded LC unit had zero/non-zero firing rates, as indicated in (**A**).

in single-unit ACC activity were not associated with differences in coordinated activity patterns of pairs of neurons in LC.

## Relationship between pupil diameter, LC spiking, and ACC coordinated activity during passive fixation

To further examine relationships between coordinated activity in ACC and activation of LC-linked arousal systems, we examined both relative to changes in pupil diameter. During passive fixation, there is a quasi-periodic fluctuation of the pupil (*Lowenstein and Loewenfeld, 1969*; *Pong and Fuchs, 2000*; *Joshi et al., 2016*). Consistent with our previous findings (*Joshi et al., 2016*), the timing of LC spiking was related to the phase of these ongoing pupil fluctuations, such that LC spiking tended to be higher preceding dilation versus constriction. These modulations showed a systematic precession with dilation phase that corresponded to a delay of ~270 ms from the maximum modulation of LC activity to the relevant pupil change (*Figure 5A*; median of intercepts from linear regression of per-time-bin peaks of individual LC PETHs; bootstrapped 95% confidence interval from regression for each LC neuron = [177–353 ms]; for examples of pupil cycles and measurement epochs, see *Figure 5—figure supplements 1 and 2*, respectively).

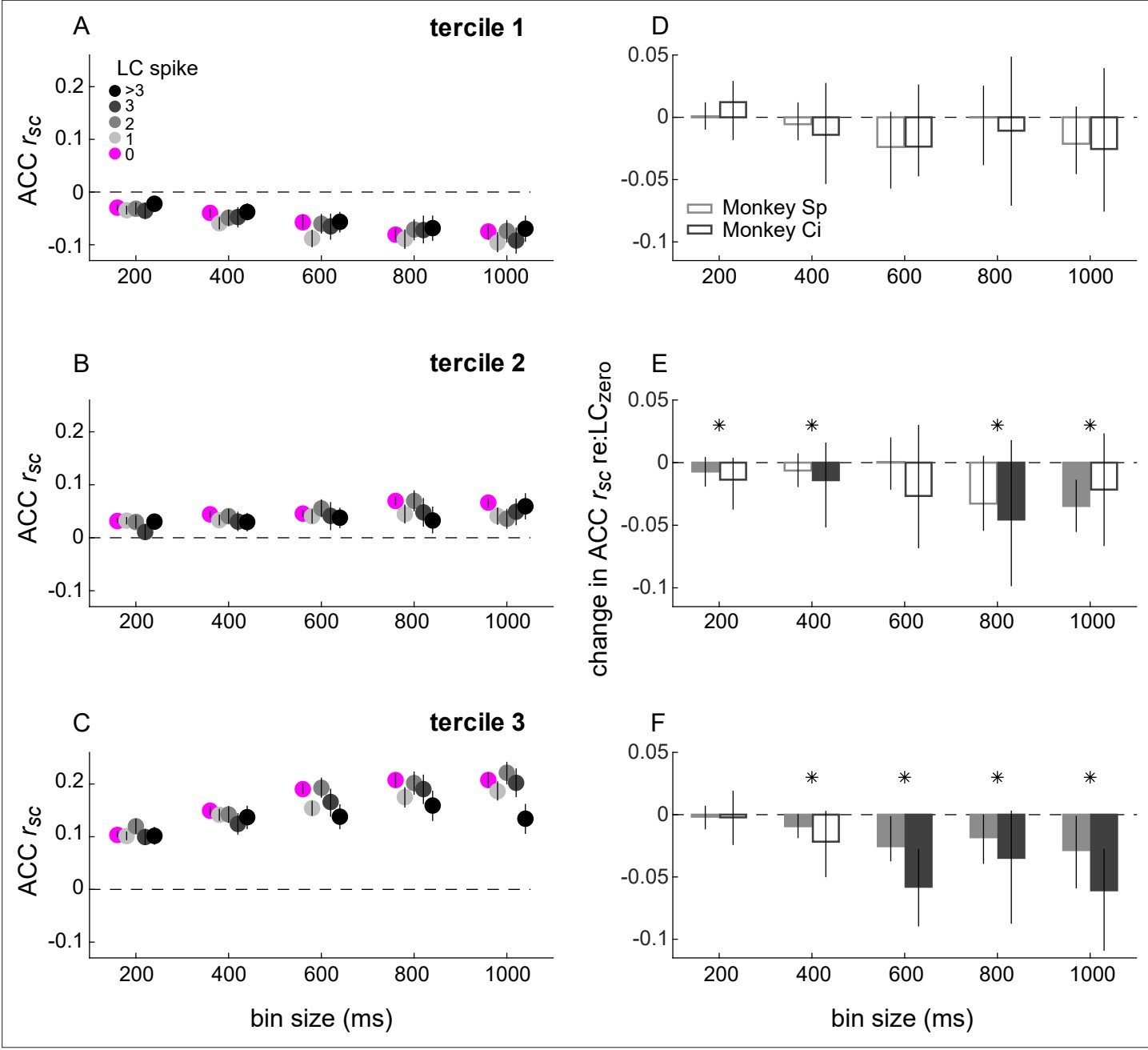

**Figure 4.** Spike-count correlations ($r_{sc}$) within anterior cingulate cortex (ACC) conditioned on simultaneously measured locus coeruleus (LC) spiking. (A–C) ACC $r_{sc}$ plotted as a function of bin size for each LC spike condition indicated in the legend in (A). The three panels separate data by ACC pairs with $r_{sc}$ values that, without reference to LC firing and for each bin size, were in the lower (A), middle (B), or upper (C) tercile from all recorded ACC pairs. Symbols and error bars are median and bootstrapped 95% confidence intervals across the given set of ACC pairs. (D–F) Difference in ACC $r_{sc}$ between the $LC_{non-zero}$ condition and the $LC_{zero}$ condition, computed for each ACC pair and plotted separately for the terciles in (A–C), respectively. Bars and error bars are median and bootstrapped 95% confidence intervals across the given set of ACC pairs. Filled bars and symbols indicate $p < 0.05$ for sign-rank tests for $H_0$: median difference between $LC_{zero}$ and $LC_{non-zero}$ conditions = 0 tested: (1) separately for each monkey (filled bars) and (2) using data combined from both monkeys (*).

The online version of this article includes the following figure supplement(s) for figure 4:

**Figure supplement 1.** Anterior cingulate cortex (ACC) $r_{sc}$ conditioned on simultaneously measured locus coeruleus (LC) spiking using shuffled trials.

**Figure supplement 2.** Relationship between locus coeruleus (LC)-linked changes in anterior cingulate cortex (ACC) spiking and ACC $r_{sc}$.

**Figure supplement 3.** Spike-count correlations ($r_{sc}$) within locus coeruleus (LC) conditioned on simultaneously measured anterior cingulate cortex (ACC) spiking.

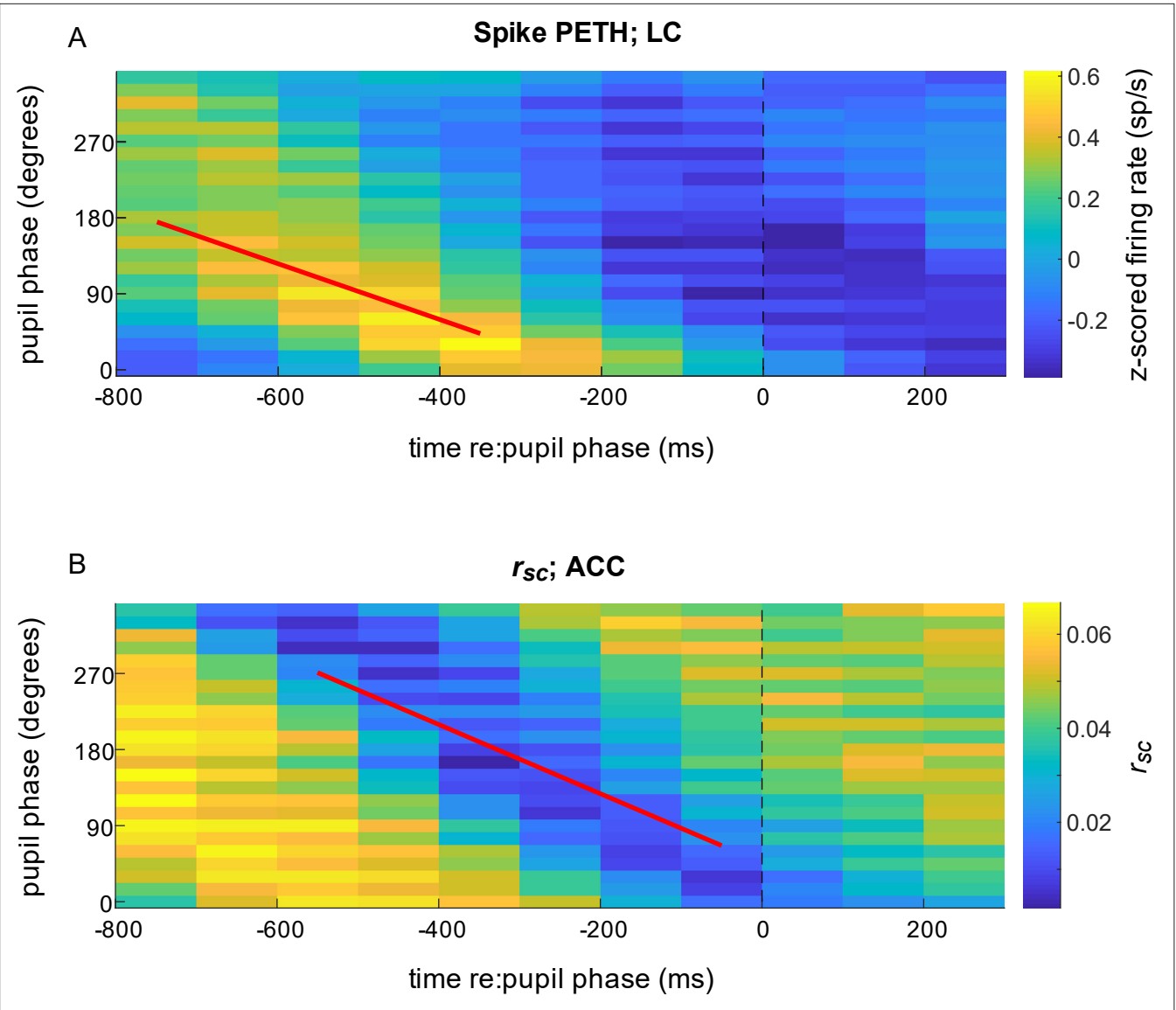

**Figure 5.** Spiking responses of locus coeruleus (LC) neurons and correlated activity in anterior cingulate cortex (ACC) relative to pupil phase. (**A**) Mean LC spike rate (colormap, in sp/s z-scored per unit) computed in 500 ms-wide bins aligned to the time of occurrence of each pupil phase. For each complete pupil cycle, phase is defined with respect to the maximum rate of dilation (0°), the maximum size (90°), the maximum rate of constriction (180°), and the minimum size (270°; see *Figure 6—figure supplement 1*). Thus, the color shown at time = *t* (abscissa), phase = *p* (ordinate) corresponds to the mean spiking rate from all LC neurons that occurred in a 500 ms-wide bin centered at *t* ms relative to the time of pupil phase *p*. Diagonal structure with a slope of ~–0.3 deg/ms implies a consistent relationship between LC firing and pupil phase for the given range of temporal offsets and pupil fluctuations, or hippus, that have a period of ~600 ms (*Joshi et al., 2016*). (**B**) ACC $r_{sc}$ aligned to pupil phase, computed as in (**A**). In both panels, data are combined from all sessions for visualization. Lines are plotted using the median regression coefficients from statistically reliable linear fits ($H_0$: slope = 0, p < 0.05) to the maxima of phase-aligned LC spiking computed per unit (**A**; median [IQR] slope = −0.33 [−0.48–0.22] deg/ms) and minima of the phase-aligned ACC $r_{sc}$ computed per ACC pair (**B**; slope = −0.41 [−0.54–0.25] deg/ms).

The online version of this article includes the following figure supplement(s) for figure 5:

**Figure supplement 1.** Pupil phase examples.

**Figure supplement 2.** Measurement of spikes relative to pupil phase.

Combined with our current findings that pairwise correlations in ACC tended to be smaller when LC is active during passive fixation, these results predicted that pairwise correlations in ACC should also vary systematically with the phase of ongoing fluctuations in pupil diameter. We found such a relationship between pupil phase and ACC $r_{sc}$ values. Specifically, over a wide range of pupil phase, decreases in ACC $r_{sc}$ followed increases in LC spiking (the fitted line in panel B is shifted to the right

relative to the fitted line in A; note also that increases in ACC $r_{sc}$, seen as the bright yellow band in the lower-left corner of panel B, occurred even earlier and are examined in more detail in Figure 9). These modulations showed a systematic progression with pupil phase over roughly the same time frame as LC firing, implying that modulations of both LC firing and ACC $r_{sc}$ had relatively fixed temporal relationships to pupil fluctuations and therefore to each other, as well (**Figure 5B**; Wilcoxon rank-sum test for $H_0$: equal LC PETH and ACC $r_{sc}$ slopes, p = 0.23).

## Single-neuron activity in response to startling events

We further examined relationships between LC and ACC neural activity in the context of external events that can cause a startle response. We showed previously that a brief, loud sound played on

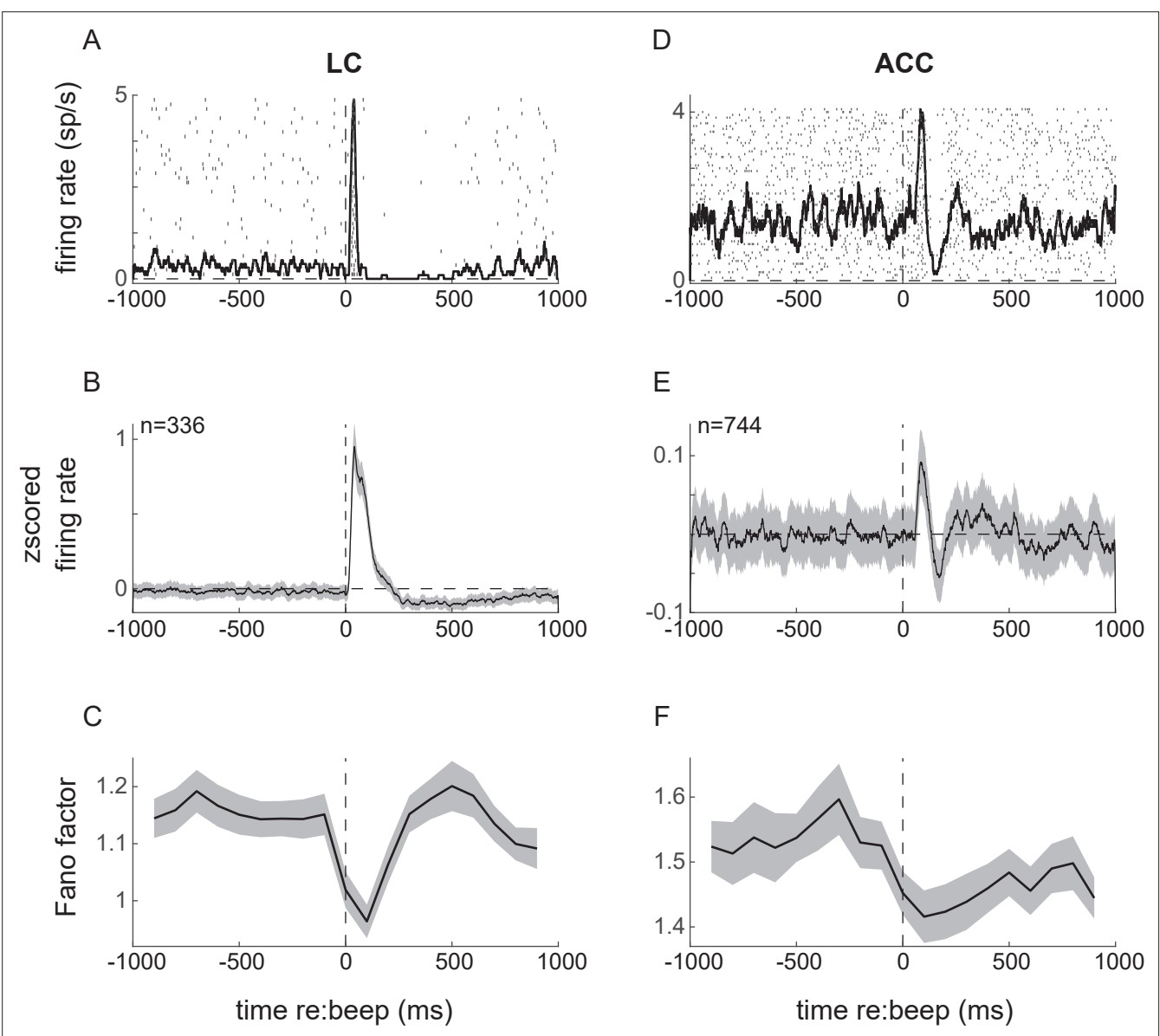

**Figure 6.** Neuronal responses to startling events (beep trials) in locus coeruleus (LC; left) and anterior cingulate cortex (ACC; right). (**A**) Example LC unit spike raster and PSTH relative to beep onset. (**B**) LC population average response. (**C**) Fano factor as a function of time relative to beep onset, calculated in 200 ms windows. (**D–F**) ACC responses relative to beep onset plotted as in (**A–C**). Lines and ribbons in (**B**), (**C**), (**E**), and (**F**) indicate mean ± sem (standard error of the mean) across all trials for all monkeys.

The online version of this article includes the following figure supplement(s) for figure 6:

**Figure supplement 1.** Anterior cingulate cortex (ACC) single-unit responses to the startle stimulus (beep).

randomly selected fixation trials ('beep trials') can elicit a transient pupil dilation as well as transient responses from individual LC neurons (*Joshi et al., 2016*; evoked responses from an example LC neuron are shown in *Figure 6A*, and the LC population average from three monkeys is shown in *Figure 6B*). The beep stimulus also elicited a consistent, albeit weaker, response in the ACC (evoked responses from an example ACC neuron are shown in *Figure 6D*, and the ACC population average from two monkeys is shown in *Figure 6E*). The averaged ACC response included a slight excitation and then inhibition relative to baseline within 200 ms of stimulus presentation. Although we found some individual cases with a biphasic response (e.g., *Figure 6D*), the average response likely resulted from the range of heterogeneous responses of individual ACC neurons (*Figure 6—figure supplement 1*).

The startle events also caused transient reductions in the trial-to-trial variability (measured as the Fano factor) of spiking activity of individual neurons in both LC and ACC, with a slightly more sustained effect in the ACC (*Figure 6C and F*). This reduction in neuronal variability following the onset of a stimulus, also referred to as quenching, has been reported previously for neurons in a range of other cortical regions and for a range of task conditions (*Churchland et al., 2010*).

## Coordinated activity in response to startling events

In the ACC, we found no evidence that the startling sound caused systematic changes in coordinated activity when measured independently of the LC response. Specifically, spike-count correlations were highly variable across ACC neuron pairs, with no statistically reliable, systematic differences when compared before versus after the beep (comparing ACC $r_{sc}$ before versus after beep for spikes measured in 1 s bins; Mann–Whitney $U$-test, $p > 0.05$ for both monkeys).

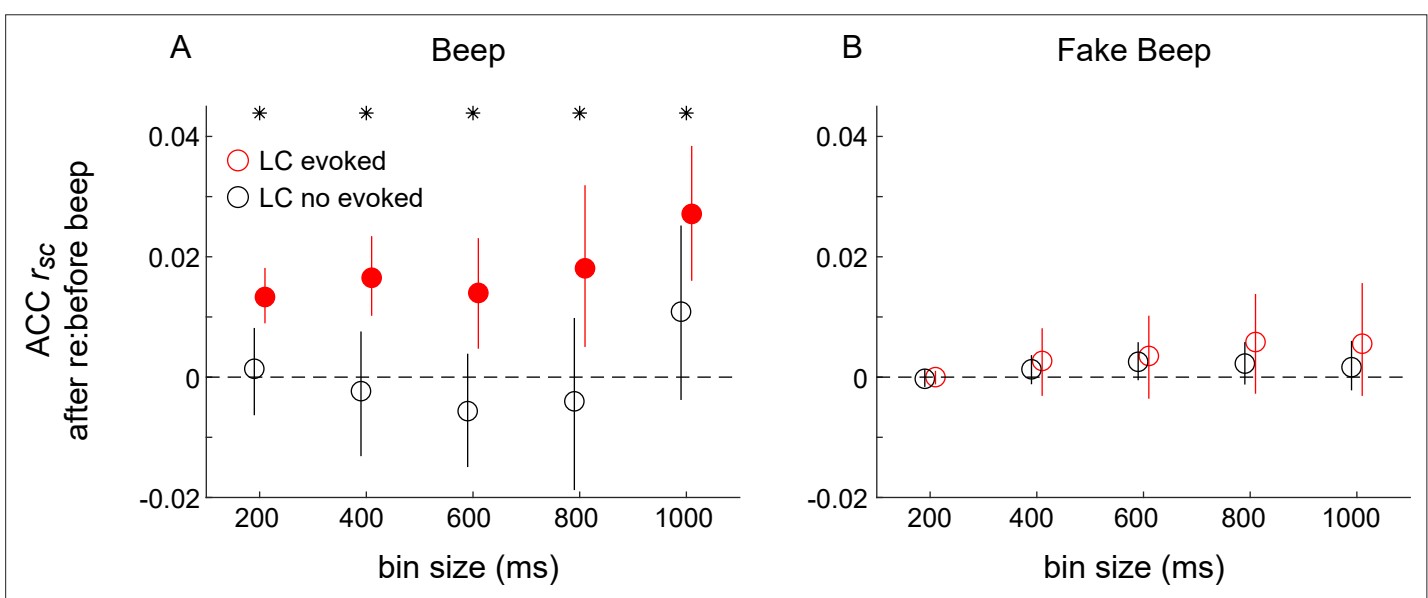

**Figure 7.** Differences in correlated activity in anterior cingulate cortex (ACC) in response to startling events, conditioned on locus coeruleus (LC) spiking. (**A**) Beep-related difference in ACC $r_{sc}$ for trials in which LC had a transient response relative to trials in which LC had no transient response, plotted as a function of the bin size used to count spikes in ACC. Circles and vertical lines are median and bootstrapped 95% confidence estimates across the given set of ACC pairs. (**B**) Data from 'fake-beep trials' (trials with no beep but sorted according to whether or not there was a transient increase in LC spiking comparable in magnitude to the beep-evoked response), plotted as in (**A**). In both panels, asterisks indicate Mann–Whitney $U$-test for $H_0$: median difference in ACC $r_{sc}$ (after relative to before the beep or 'fake-beep') between the two groups (LC-evoked and no-evoked) is different for the given time bin, $p < 0.05$ for both monkeys' data pooled together; filled circles indicate sign-rank test for $H_0$: ACC $r_{sc}$ differences (after relative to before the beep) within each group is different from zero, $p < 0.05$ for both monkeys' data pooled together.

The online version of this article includes the following figure supplement(s) for figure 7:

**Figure supplement 1.** Consistency of locus coeruleus (LC) responses to startling sounds.

**Figure supplement 2.** Differences in anterior cingulate cortex (ACC) Fano factor in response to startling events, conditioned on locus coeruleus (LC) spiking.

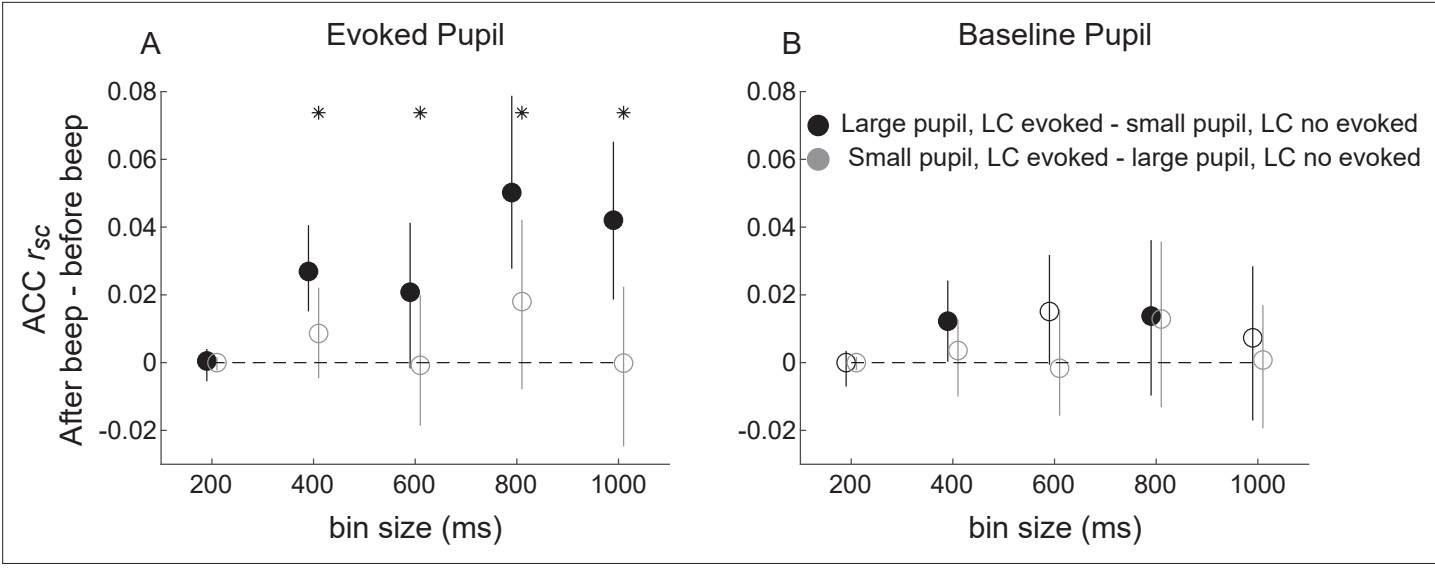

**Figure 8.** Beep-related differences in anterior cingulate cortex (ACC) correlations relative to locus coeruleus (LC) responses and pupil size. (**A**) Difference in ACC $r_{sc}$ computed after versus before the beep plotted as a function of bin size for trials groups based on evoked LC spiking and evoked pupil dilations (groups as indicated in the legend in **B**). (**B**) Difference in ACC $r_{sc}$ computed after versus before the beep plotted as a function of bin size for trials groups based on evoked LC and baseline pupil size. Circles and vertical lines are median and bootstrapped 95% confidence estimates across the given set of ACC pairs. In both panels, asterisks indicate Mann–Whitney $U$-test for $H_0$: median difference in ACC $r_{sc}$ (after relative to before the beep) is different for trials with (LC-evoked) versus without (LC not evoked) a transient LC response for the given bin size, $p < 0.05$ for both monkeys' data pooled together; filled circles indicate sign-rank test for $H_0$: ACC $r_{sc}$ differences (after relative to before the beep) within each group is different from zero, $p < 0.05$ for both monkeys' data pooled together. An ANOVA with groups (black and gray symbols as indicated in **B**), bin size, and pupil measure (baseline or evoked) as factors showed reliable effects of group and the interaction between group and pupil measure.

The online version of this article includes the following figure supplement(s) for figure 8:

**Figure supplement 1.** Relationship between startling sound-driven changes in pupil diameter and locus coeruleus (LC)-evoked activation.

In contrast, we found systematic relationships between the beep-induced changes in coordinated activity in the ACC and the simultaneously measured LC response. Specifically, LC neurons did not show a characteristic transient response on every presentation of a startle stimulus: the median fraction of trials with a response that included a transient increase and then decrease in spiking activity was 54% for monkey Sp and 43% for monkey Ci (*Figure 7—figure supplement 1*). Therefore, we tested if and how arousal-linked changes in ACC correlated activity depended on whether or not the simultaneously recorded LC neuron responded transiently to the startling sound. Specifically, we computed ACC $r_{sc}$ in time bins of different sizes, separately for 1 s preceding and 1 s following the sound. ACC $r_{sc}$ tended to be larger after versus before the beep stimulus, but only for the subset of trials in which the beep also elicited the characteristic LC response (*Figure 6A and B*), for both monkeys (*Figure 7A*). These modulations did not reflect simply transient differences in LC firing rates but rather specific differences between the presence and absence of the stimulus-evoked LC response (*Figure 7B*). These modulations of ACC pairwise correlations also did not reflect LC-driven differences in the stimulus-driven quenching of response variability in ACC, which was similar on trials with and without LC-evoked responses (*Figure 7—figure supplement 2*).

## Relationship between pupil diameter, LC spiking, and ACC coordinated activity in response to startling events

We showed previously that the magnitude of beep-evoked LC responses is correlated positively with the size of the simultaneously evoked change in pupil diameter (*Joshi et al., 2016*). We reproduced those results in our current data set (*Figure 8—figure supplement 1*). We extended those results by showing that these pupil-LC relationships also relate systematically to beep-evoked changes in ACC $r_{sc}$. Specifically, we measured beep-evoked changes in ACC $r_{sc}$ separately for trials grouped by LC and pupil responses, focusing on how large or small pupil responses related to the evoked-LC versus no-evoked-LC differences in ACC $r_{sc}$ shown in *Figure 7*. In general, the beep-evoked changes in ACC

$r_{sc}$ that required an LC-evoked response were larger on trials in which the evoked LC response was accompanied by a large versus small pupil dilation (*Figure 8A*). These effects were not evident when considering baseline, not evoked, pupil size (*Figure 8B*).

## Relative timing of neural activity patterns in LC and ACC

To better understand the relative timing of firing patterns (both individual neuron firing rate changes and paired-neuron $r_{sc}$ changes) in LC and ACC, we measured their time courses relative to two external events with well-defined timing: onset of stable fixation on no-beep trials and onset of the sound on beep trials (*Figure 9*). For no-beep trials, LC firing rates increased after the fixation point turned on and before stable fixation was acquired (*Figure 9A*). This epoch also included elevated ACC $r_{sc}$ (particularly when the simultaneously recorded LC neuron was not active, consistent with the results shown in *Figure 4*) that preceded the elevation in LC firing (*Figure 9B*, gray lines). Both LC firing and ACC $r_{sc}$ tended to stabilize after stable fixation was attained. These results are consistent with the possibility that LC activation prior to fixation onset drives (or at least coincides with) a lasting decrease in ACC $r_{sc}$ during the remainder of the trial.

Following a beep, there was a different pattern of relationships between LC firing and ACC $r_{sc}$. Specifically, we divided trials that did (*Figure 9C and D*) or did not (*Figure 9E and F*) have a beep-evoked LC response (like in *Figures 7 and 8*). In general, the beep caused a transient increase in ACC $r_{sc}$, and this increase tended to occur more rapidly on trials with (*Figure 9D*, gold shaded area) versus without (*Figure 9F*, gold shaded area) a concomitant LC response (Mann–Whitney $U$-test, for $H_0$: the linear rate of increase of ACC $r_{sc}$ in this period was larger when LC neuron did versus did not respond, p = 0.0297). Note that the LC transient response (*Figure 9C*) peaked at the beginning of the epoch in which the ACC $r_{sc}$ started to increase, which is consistent with the idea that the LC-evoked response drives (or at least coincides with) a faster rate of ACC $r_{sc}$ increase (steeper slope in the gold region in *Figure 9D* compared with F). The evoked LC response was also associated with a more reliable and larger increase in ACC $r_{sc}$ later in the trial (compare *Figure 9D and F*; Mann–Whitney $U$-test for $H_0$: ACC $r_{sc}$ 500–1000 ms after the beep was the same on trials with versus without a beep-evoked LC response, p = 0.0350 for both monkeys' data pooled together) but was not associated with differences in ACC $r_{sc}$ in the time just before the beep (i.e., ACC $r_{sc}$ did not anticipate whether or not the beep would evoke an LC response; Mann–Whitney $U$-test for $H_0$: ACC $r_{sc}$ –500–0 ms before the beep was the same on trials with versus without a beep-evoked LC response, p = 0.0693 for both monkeys' data pooled together).

## Discussion

We measured relationships between pupil-linked activation of the LC-NE system and changes in cortical neural activity patterns. Previously, we showed that changes in pupil size during passive, near fixation and when driven by external startling events covary with the timing of spiking activity in LC and parts of brainstem and cortex, including the ACC (*Joshi et al., 2016*). Other studies have shown that pupil size and other indirect measures of LC activation can also correspond to changes in coordinated activity in sensory cortex under certain conditions (*Reimer et al., 2014*; *Reimer et al., 2016*; *Vinck et al., 2015*; *McGinley et al., 2015*). These previous studies did not explore the relationships between ongoing versus evoked LC activation (or NE release) and concurrent changes in both coordinated cortical activity and pupil size. We extended those findings, including one study that elicited changes in coordinated activity in cortex via manipulation of LC activity (*Devilbiss and Waterhouse, 2011*), by showing that LC neuronal activity is reliably associated with changes in correlated spiking activity (spike-count correlations, or $r_{sc}$) between pairs of neurons in ACC. We further showed that these LC-linked changes in ACC $r_{sc}$ depend on the nature of the LC activation. During passive fixation and in the absence of controlled external inputs, ongoing LC firing was associated with a reduction in ACC $r_{sc}$. In contrast, startling sounds drove transient increases in LC firing that were associated with an increase in ACC $r_{sc}$. Under both conditions, LC and pupil-linked changes in ACC $r_{sc}$ were most pronounced over relatively long time windows (>500 ms) that are consistent with neuromodulatory timescales and thus might involve the effects of LC-mediated NE release in the ACC (*Feldman RS et al., 1997*; *McCormick and Prince, 1988*; *Wang and McCormick, 1993*; *Schmidt et al., 2013*; *Timmons et al., 2004*).

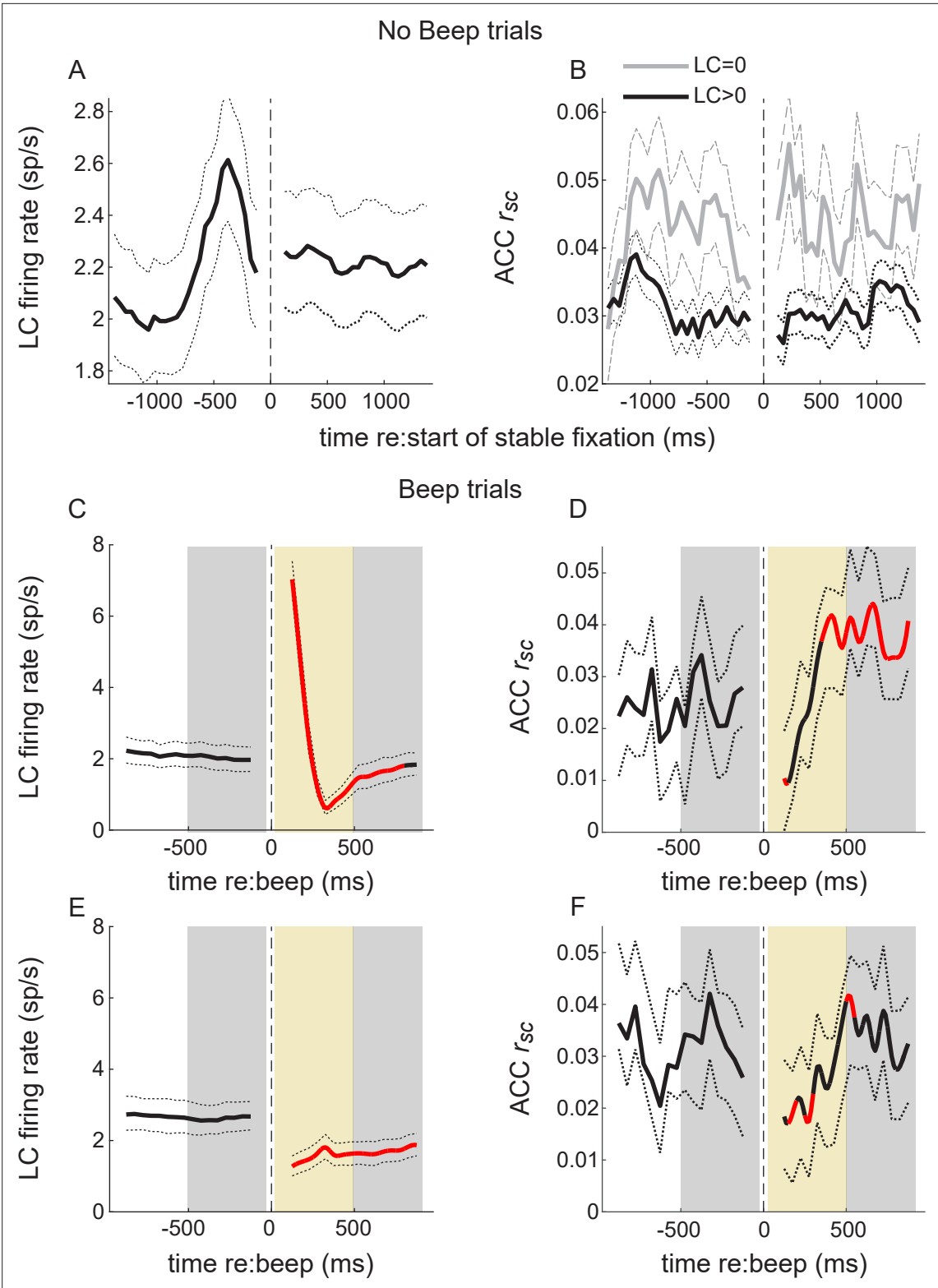

**Figure 9.** Temporal relationships between locus coeruleus (LC) firing and anterior cingulate cortex (ACC) correlated activity. No-beep trials: (**A**) LC firing rate aligned to the start of stable fixation. Solid and dotted lines are mean and bootstrapped 95% confidence estimates across the set of LC neurons. (**B**) Mean and 95% confidence estimates of ACC $r_{sc}$ computed across trials in time bins aligned to the start of stable fixation, separated into trials on which the simultaneously measured LC neuron did not (LC = 0) or did (LC > 0) fire an action potential. Beep trials (**C–F**). Panels as in (**A**) and (**B**), except with LC firing and ACC $r_{sc}$ now aligned to the time of the beep and separated into trials with (**C, D**) or without (**E, F**) an LC response to the beep.

*Figure 9 continued on next page*

*Figure 9 continued*

Red portions of lines indicate Mann–Whitney *U*-test for $H_0$: per bin value is different from pre-beep baseline, p < 0.05 for both monkeys' data pooled together. Gray shaded areas in (**C**–**F**) indicate epochs in which statistical comparisons were made for trials with (LC-evoked) versus without (LC not evoked) an LC transient response; gold shaded areas indicate epochs in which ACC $r_{sc}$ slopes were compared. For trials, see text.

The context dependence of these effects implies that spontaneous versus sensory-evoked activation of the LC-NE system can have different effects on coordinated patterns of activity in cortical networks. These effects seem likely to involve the different LC firing patterns under these conditions, which for sensory-evoked activity include not just a large, transient increase in spike rate but also a reduction in spike-count variability and heterogenous relationships to pupil-linked changes in arousal (*Joshi et al., 2016*). In principle, different temporal patterns of NE release could mediate a range of synaptic effects via distinct NE receptor subtypes with different affinities and different spatial distributions in different parts of the brain (*Aoki et al., 1987*; *Nicholas et al., 1993*; *Arnsten et al., 1998*; *Arnsten, 2000*; *Berridge et al., 2012*; *Berridge and Spencer, 2016*). These synaptic effects can influence both excitatory and inhibitory neurons as well as astrocytes, resulting in changes in network dynamics in thalamus, cortex, and elsewhere (*Segal and Bloom, 1976*; *Dillier et al., 1978*; *Waterhouse et al., 1980*; *McCormick and Prince, 1988*; *Wang and McCormick, 1993*; *McLean and Waterhouse, 1994*; *Fernández-Pastor and Meana, 2002*; *Salgado et al., 2011*; *Salgado et al., 2016*; *Paukert et al., 2014*; *Guan et al., 2015*; *Schiemann et al., 2015*; *Sherpa et al., 2016*; *Garcia-Junco-Clemente et al., 2019*; *Aston-Jones and Cohen, 2005*; *Ohshima et al., 2017*; *Rodenkirch et al., 2019*). Exactly how these diverse mechanisms support context-dependent changes in coordinated activity patterns merits further study, perhaps by selectively controlling the magnitude and timing of NE release in cortex while assessing changes in coordinated activity and using biophysically inspired models to test hypotheses about the underlying mechanisms (*de la Rocha et al., 2007*; *Doiron et al., 2016*).

In contrast to the relationships that we identified between LC activation and coordinated ACC activity patterns, we did not find similarly reliable relationships between the spiking activity of individual neurons measured simultaneously in each of the two brain regions. For example, we found slight increases and decreases in mean firing rates and a marked reduction in the trial-to-trial variability of firing rates of individual units in the ACC in response to external stimuli, as has been reported previously for other cortical regions (*Churchland et al., 2010*). However, these changes in single-unit ACC responses were not related reliably to properties of the concurrently measured LC response that we could measure and thus may involve mechanisms other than the LC-NE system. These findings do not appear to be consistent with previous work showing improvements in signal-to-noise ratios in cortex in response to LC-NE activation or during states of higher versus lower global arousal (*Kolta et al., 1987*; *McLean and Waterhouse, 1994*; *Lee et al., 2018*; *Lombardo et al., 2018*). This difference could reflect differences in the specific neurons that we targeted in ACC, the different task conditions we tested relative to previous studies, analyses based on ongoing versus evoked LC activity, or some combination of these factors, and merits further study.

What might be the function of LC- and arousal-linked changes in ACC correlations? In general, correlations in spiking activity can be useful or detrimental depending on a number of factors, many of which remain unexplored, particularly outside of sensory cortex (*Averbeck et al., 2006*; *Cohen and Kohn, 2011*; *Kohn et al., 2016*). For example, an increase in correlations is considered a possible mechanism for connecting neural populations over a range of spatial and temporal scales (*Gray et al., 1989*; *Singer, 1999*; *Riehle et al., 1997*). Some computational schemes can benefit from increases in correlated variability (*Abbott and Dayan, 1999*; *Singer and Gray, 1995*; *Gray, 1999*; *Kohn et al., 2016*; *Valente et al., 2021*). However, an increase in correlations can also negatively impact the information coding capacity of a large neural population, particularly over longer integration windows (*Zohary et al., 1994*; *Bair et al., 2001*; *Averbeck et al., 2006*; *Renart et al., 2010*; but also see *Nirenberg and Latham, 2003 Moreno-Bote et al., 2014*, for alternative interpretations). Global states such as arousal and attention (traditionally linked with LC-NE and cholinergic systems, respectively) can modulate cortical correlations (*Cohen and Maunsell, 2009*; *Mitchell et al., 2009*; *Herrero et al., 2013*; *Schmidt et al., 2013*). Some work has also suggested that correlations induced by common inputs must be actively decorrelated by the action of local recurrent excitation and inhibition to preserve information fidelity (*Ecker et al., 2010*; *Renart et al., 2010*).

Our results suggest that both increases and decreases in coordinated neural activity patterns in cortex may be under context-dependent, moment-by-moment control of the LC-NE system. Event-driven transient activation of LC could help to synchronize populations of cortical neurons by shifting them to a more correlated state. Conversely, during nonevoked, ongoing (event-independent) firing, the NE release could enhance information processing and signal-to-noise ratios by reducing cortical correlations. Further work is needed to identify if and how NE-mediated changes in network dynamics subserve these functions, particularly in the context of goal-directed behaviors that involve multiple brain regions, including LC and ACC, as well as other neuromodulator systems (*Hayden et al., 2011*; *Varazzani et al., 2015*; *Ebitz and Platt, 2015*; *Alamia et al., 2019*).

## Materials and methods

Three adult male rhesus monkeys (*Macaca mulatta*) were used for this study (monkeys Oz, Ci, and Sp). All training, surgery, and experimental procedures were performed in accordance with the NIH's Guide for the Care of Use of Laboratory Animals and were approved by the University of Pennsylvania Institutional Animal Care and Use Committee (protocol 806027). The behavioral task and pupillometry recording and analysis techniques were identical to those we used previously (*Joshi et al., 2016*). Briefly, fixation trials were of variable length (1–5 s, uniformly distributed). The monkey was rewarded with a drop of water or diluted Kool-Aid for maintaining fixation until the end of the trial. On a subset of randomly chosen trials (~25%), after 1–1.5 s of fixation a sound (1 kHz, 0.5 s) was played over a speaker in the experimental booth ('beep trials'). The monkey was required to maintain fixation through the presentation of the sound, until the fixation point was turned off.

### Electrophysiology

Each monkey was implanted with a recording cylinder that provided access to LC+ (the LC and adjacent, NE-containing subcoeruleus nucleus; *Sharma et al., 2010*; *Paxinos et al., 2008*; *Kalwani et al., 2014*), inferior colliculus (IC), and superior colliculus (SC). The detailed methodology for targeting and surgically implanting the recording cylinder and then targeting, identifying, and confirming recording sites in these three brain regions is described in detail elsewhere (*Kalwani et al., 2014*; *Joshi et al., 2016*). Briefly, the LC was targeted initially using custom (*Kalwani et al., 2009*) and/or commercial software (Brainsight). Tracks were then refined using electrophysiological recordings and microstimulation in brain regions dorsal to LC. Neurons in the intermediate layers of SC (SC_i) exhibited spatial tuning on a visually guided saccade task and could elicit saccades via electrical microstimulation (*Robinson, 1972*; *Sparks and Nelson, 1987*). IC units exhibited clear responses to auditory stimuli. Activity in the trigeminal mesencephalic tract (me5), located immediately dorsal to the LC, showed distinct activity related to orofacial movements such as sipping. LC+ units had relatively long action potential waveforms, were sensitive to arousing external stimuli (e.g., door knocking), and decreased firing when the monkey was drowsy (e.g., eyelids drooped; *Aston-Jones et al., 1994*; *Bouret and Sara, 2004*; *Bouret and Richmond, 2009*). Nonoptimal tracks also helped with mapping; for example, a more medial track could miss me5 but lead to the trochlear decussation with characteristic ramp-and-hold activity related to downward saccades. Likewise, tracks that encountered IC but not SC_i were likely too lateral and often missed me5 and LC. Sites were verified using MRI and assessing the effects of systemic injection of clonidine on LC+ responses in monkeys Oz and Ci and by histology with electrolytic lesions and electrode-tract reconstruction in monkey Oz (*Kalwani et al., 2014*). Recording and microstimulation in these brainstem-targeting tracks were conducted using custom-made electrodes (made from quartz-coated platinum-tungsten stock wire from Thomas Recording) and a Multichannel Acquisition Processor (Plexon, Inc).

ACC cylinders were placed at Horsley–Clarke coordinates 33 mm anterior-posterior (AP), 8 mm lateral (L), in the left hemisphere for monkey Sp and in the right hemisphere for monkey Ci. For ACC recordings, we targeted the dorsal bank of the anterior cingulate sulcus, ~4–6 mm below the cortical surface. ACC tracks were planned and refined using MRI and Brainsight software, as well as by listening for characteristic patterns of white and gray matter during recordings in an initial series of mapping experiments. Recordings were conducted using either custom-made single electrodes or multicontact linear electrode arrays (8- and 16-channel V-probe, Plexon).

For each brain region, we recorded and analyzed data from all stable, well-isolated units that we encountered. Neural recordings were filtered between 100 Hz and 8 kHz for spikes (Plexon MAP). Spikes were sorted offline (Plexon offline sorter). Electrical microstimulation in $SC_i$ consisted of biphasic (negative-positive) pulses, 0.3 ms long, 100 ms in duration, and delivered at 300 Hz via a Grass S-88 stimulator through a pair of constant-current stimulus isolation units (Grass SIU6) that were linked together to generate the biphasic pulse.

## Data analysis

For each recorded neuron, we considered spiking activity only during stable fixation, defined as a 1.1 s window that began 1 s after attaining fixation in which the monkey's gaze remained within a square window 0.2° per side centered on the fixation point. To assess ACC spiking activity patterns conditioned on LC activation, we divided trials within each session based on whether the single LC neuron from which recordings were being made either did ($LC_{non-zero}$) or did not ($LC_{zero}$) produce at least one action potential during stable fixation. We also divided session trials into four additional groups of trials in which the LC neuron fired 1, 2, 3, and ≥4 spikes.

We used 10 bin sizes ranging from 100 ms to 1 s, spaced logarithmically, to count spikes. The mean spike count, variance of spike counts, and the Fano factor (the ratio of the variance to the mean of spike counts) for each neuron were calculated across trials for each bin size. Pairwise spike-count correlations ($r_{sc}$) were calculated for each bin size as follows. First, trial spike counts from each neuron in the pair were z-scored, and trials on which the response of either neuron was >3 standard deviations different from its mean were removed to avoid effects of outlier responses (*Kohn and Smith, 2005*; *Smith and Kohn, 2008*). Then, the MATLAB function corrcoef was used to obtain the Pearson correlation coefficient. Shuffled estimates were made by calculating $r_{sc}$ from pairs of neurons with spike-count vectors generated from randomly selected trials for each neuron.

ACC $r_{sc}$ values were calculated for all trials for each pair of well-isolated neurons, independently of spiking in LC and also separately for trials divided into groups depending on spiking in LC ($LC_{zero}$ and the five LC > 0 groups). The LC-independent $r_{sc}$ values collected from all pairs and measured using ACC spikes counted using each time bin were divided into terciles that corresponded broadly to pairs that were negatively correlated (tercile 1), uncorrelated (tercile 2), or positively correlated (tercile 3). This analysis allowed us to assess whether changes in $r_{sc}$ in one region associated with spiking in the other region depended on the pairs being correlated to begin with or not.

## Data Availability Statement

Data and Matlab code for all figures in this manuscript are available at here copy archived at swh:1:rev:22b8c94380c057201fb1fcc56c6037b70c56021c; *Joshi and Gold, 2021*.

## Acknowledgements

We thank Long Ding, Adrian Radillo, Alice Dallstream, Kyra Schapiro, David Kleinfeld, and Takahiro Doi for valuable comments, Rishi Kalwani for piloting the LC/fixation studies, and Jean Zweigle for expert animal care and training. This study was funded by R21 MH107001.

## Additional information

### Competing interests

Joshua I Gold: Senior editor, *eLife*. The other author declares that no competing interests exist.

### Funding

| Funder | Grant reference number | Author |
| --- | --- | --- |
| National Institutes of Health | R21 MH107001 | Joshua I Gold |

The funders had no role in study design, data collection and interpretation, or the decision to submit the work for publication.

## Author contributions
Siddhartha Joshi, Conceptualization, Data curation, Formal analysis, Investigation, Methodology, Resources, Validation, Visualization, Writing - original draft, Writing - review and editing; Joshua I Gold, Conceptualization, Data curation, Formal analysis, Funding acquisition, Methodology, Project administration, Resources, Software, Supervision, Validation, Visualization, Writing - original draft, Writing - review and editing

## Author ORCIDs
Siddhartha Joshi ⓘ http://orcid.org/0000-0002-0529-9430
Joshua I Gold ⓘ http://orcid.org/0000-0002-6018-0483

## Ethics
All animal training, surgery and experimental procedures were performed in accordance with the NIH's Guide for the Care and Use of Laboratory Animals and were approved by the University of Pennsylvania Institutional Animal Care and Use Committee (protocol 806027).

## Decision letter and Author response
Decision letter https://doi.org/10.7554/eLife.63490.sa1
Author response https://doi.org/10.7554/eLife.63490.sa2

# Additional files

## Supplementary files
• Transparent reporting form

## Data availability
Data and Matlab code for all figures in this manuscript are available at: https://github.com/TheGoldLab/LC_ACC_paper_Joshi_Gold_2021 (copy archived at swh:1:rev:22b8c94380c057201fb1fcc56c6037b70c56021c).

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
