## [Editor Report]

This is a timely and important study that systematically assesses the relationships between neuronal activity in the locus coeruleus (LC) and the anterior cingulate cortex (ACC) in non-human primates. The LC is a major source of cortical norepinephrine that has reciprocal connectivity with the ACC, and the authors have convincingly shown that LC spiking is associated with changes in ACC spike correlations. Further, these changes have consistent phase relationships with pupil size. This is a rare data set that is technically challenging to acquire, and the results are an important advance toward understanding a circuit that is likely to play a role in regulating brain states such as arousal or attention.

---

## [Decision Letter]

**Decision letter after peer review:**

Thank you for submitting your article "Relationships between Locus Coeruleus Firing Patterns and Coordinated Neural Activity in the Anterior Cingulate Cortex" for consideration by *eLife*. Your article has been reviewed by 3 peer reviewers, and the evaluation has been overseen by a Reviewing Editor and Tirin Moore as the Senior Editor. The following individual involved in review of your submission has agreed to reveal their identity: Tobias H Donner (Reviewer #1).

The reviewers have discussed the reviews with one another and the Reviewing Editor has drafted this decision to help you prepare a revised submission.

We would like to draw your attention to changes in our revision policy that we have made in response to COVID-19 (https://elifesciences.org/articles/57162). Specifically, we are asking editors to accept without delay manuscripts, like yours, that they judge can stand as *eLife* papers without additional data, even if they feel that they would make the manuscript stronger. Thus the revisions requested below primarily address clarity and presentation.

Summary:

This is a monkey neurophysiology study into the neuronal basis of arousal in the primate brain. The authors systematically assess the relationship between neuronal activity in an important neuromodulatory center of the brainstem, the locus coeruleus (LC), and a reciprocally connected cortical region, the anterior cingulate cortex (ACC). LC is a major source of cortical norepinephrine (NE), so LC spikes may predict momentary changes in cortical NE. Pupil size, also measured here, is sometimes used as a peripheral index of NE levels in the cortex, though this is also correlated with a variety of other factors, including other neuromodulators. The authors have three main conclusions. First, spikes in LC neurons predicted a decrease in the Fano factor in ACC and a decrease in the pair-wise correlations (r_sc_) between highly correlated ACC neurons. Second, both LC spikes and ACC r_sc_ appeared to have a consistent phase relationship with pupil size, with the troughs in ACC r_sc_ lagging LC spikes. Third, LC spikes predicted changes in the relationship between surprising stimuli and ACC r_sc_, as well as the relationship between the pupil response to surprising stimuli and ACC r_sc_. The authors also mention that these changes are independent of the relationship of LC activity and ACC firing rates.

Overall, the reviewers felt that this is a timely and important study, particularly because the LC-ACC circuit is under-characterized in primates. The major strengths of the study include the rarity of the data set, the technical sophistication of the analyses, and the investigation of LC-ACC relationships across multiple timescales. However, it was felt that revisions are needed to make some points convincingly, and in other cases the inherent limitations in the analyses should be more thoroughly acknowledged. The specific comments on these points are outlined below.

Essential revisions:

1. The analyses in this study do not directly assess causality or directionality of the interactions reported. This was noted my multiple reviewers, with specific points in the comments below. In addition to these points, in light of the difficulty in claiming causality or directionality from recording data, and it was agreed that there should be a general restructuring of the interpretation to reflect this limitation.

a. While the data are very interesting and compelling in their current form. I think that some limitations of the current analyses should be acknowledged. Specifically, the analyses do not allow for inferences about the directionality of effects. So, it remains open if the changes of LC firing rate cause the changes in spike-count correlations in ACC (as seems likely), or vice-versa; or if a third variable causes the effects in both brain regions? Even without causal manipulations, inferences of this kind could be based on an assessment of the temporal relationships of changes in the local signal properties. It could also be based on statistical assessments (e.g. using multivariate autoregressive modeling) of "Granger causality".

b. Related to "Single-neuron activity during passive fixation"/Figure 2. I'm curious to understand the direction of this effect – does variance in ACC predict spike counts in LC or do spikes in LC predict variance in ACC? Is it possible to look at spike-evoked Fano factor in the ACC (i.e. before and after an LC spike)? Figure 6 is described as implying that these LC spikes and ACC r_sc_ are temporally related, but this is only analyzed as mediated by the relationship between each and pupil size, but this temporal relationship appears not to be investigated directly ("relatively fixed temporal relationships to pupil fluctuations and therefore to each other").

c. The authors do nice internal controls testing not only LC-ACC effects, but also ACC-LC effects. They describe that LC-ACC are significant, while the ACC-LC effects are less reliable. This is important for their claims but also as a validation of the analyses. Did the authors formally compare whether there are significant differences between ACC-LC and LC-ACC effects? Showing that ACC-LC is not significant does not address this per se. Slopes (Figure 5) can definitely be tested for LC-ACC vs ACC-LC. (e.g., is the slope more negative in one versus the other).

d. Figure 6A, in the legend is described as evidence that LC spikes have a consistent phase relationship with pupil fluctuations that have a period of 600 ms. In the text, this result is taken as evidence that LC peaks 270 ms in advance of the "relevant pupil change" (I'm not entirely clear what pupil phase is being referenced by this phrase). Are these different interpretations? Do LC spikes have a fixed relationship to one component of the pupil fluctuations (like dilation or the cresting at peak size) or are they entrained to the oscillation? Also, it would be good to cite Pong and Fuchs 2000 J Neurophysiol in addition to Joshi, 2016 as evidence for hippus in the monkey at this 1.67 Hz frequency.

e. Related, in Figure 6B, it looks (to my eye) like ACC r_sc_ is also peaking in advance of the trough highlighted in this figure. This would suggest an alternative model, where high r_sc_ in ACC predicts spontaneous LC spiking, and then lower r_sc_ in ACC. This alternative might more in line with Alla Karpova's work that focuses on the effects of ACC on LC activity, rather than the LC-ACC relationship that is the focus of this paper.

2. A major caution in interpreting these results is that the paper performs a lot of multiple comparisons in nested bins, and it is not clear that the multiple comparison problem is appropriately controlled for. Many effects appear obvious in the plots suggesting that the three major results would survive correction (i.e. Figure 2G), but for some of the latter analyses it is not clear that effects would survive correction. This problem is complicated by the fact that the tests are strongly interdependent, so it's not clear that a simple correction would be sufficient. It may be more appropriate to conduct permutation tests, or directly ask how independent variables alter how the dependent variable scales with bin sizes.

3. There was some confusion surrounding the motivation for the study as stated in the Introduction. Specifically, the tonic/phasic dichotomy is mentioned in the introduction, but it appears that this was not directly investigated in the rest of the manuscript. Moreover, it was noted that this dichotomy may not be so clear-cut. There may be some overlap with the ideas of ongoing/evoked activity, as in the Aston-Jones and Cohen (2005) framework, however the links between these perspectives are not explored. Reviewers recommend either describing in more detail how the tonic/phasic perspectives motivated the study, or removing this from the introduction and better explaining why one might be curious about the relationship between LC spikes and ACC activity. In the same vein, it would also be helpful to motivate the specific analyses that were performed. Later analyses were well motivated, but the rationale behind the first few were less clear.

4. The authors claim that the firing rate changes in ACC are not reliably related to LC activity, yet the effects appear significant when monkeys are combined? As the authors know, short times scale firing rate correlations (and cross correlations) even in anatomically connected areas are very hard to detect but nonetheless could be real. I think the authors need to take these effects into account to make a convincing case that the other effects they focus on cannot be explained by small population level shifts in rate (or also particularly in fano factor which is highly relevant to population level correlations). I believe this is in their data but it should be fleshed out and the above points should be discussed.

5. Two conceptual points were raised that should be addressed in the discussion or elsewhere:

a. It is reported that LC phasic activity related to bottom up salient cues (e.g. "surprise" or "startle") increases correlations in ACC, so does this mean the ACC is able to encode less information during surprising events as would be predicted by most theories of correlated activity and information coding in cortex? This has interesting implications for ACC functions, if true.

b. The current literature on LC is very preliminary and theoretic. Precisely, while strong assumptions exist about what it encodes, we don't know if it encodes RPEs, surprises, intense sensory events, higher order RPEs (e.g. some belief state violation related signal), etc. It looks like it is likely complex, however the present task is very simple and may miss important nuances in the LC-ACC network. It is important to point this out and explicitly indicate early in the paper that the procedure is meant to "elicit" LC firing states, rather than test what Lc encodes.

6. It was concluded that LC activity has a context-dependent effect on ACC r_sc_, increasing it during passive viewing, but not changing it with LC activity is evoked by a surprising event. However, this conclusion is based on comparing post-stimulus ACC r_sc_ to the pre-stimulus baseline, which does not rule out the alternative interpretation that ACC activity before a surprising event predicts the likelihood of an LC spike. To elaborate, it seems like there is largely a change in the pre-beep, baseline ACC r_sc_ in this data. This would imply that elevated ACC r_sc_ before beep trials predicts no phasic response in LC. If so, the decrease in ACC r_sc_ after LC might be a simple homeostatic effect (i.e. due to the tendency to return to baseline), rather than a context-dependent effect of LC spikes on ACC r_sc_. Did the authors consider this alternative model? Further, the paper does not show that the pattern of ACC r_sc_ after a surprising beep is any different following an LC spike than it is in the absence of an LC spike.

7. In Figure 8, it's not clear if the effects are due to the fact that beep-evoked changes are happening over longer time scales or if they're happening at different latencies relative to the beep (i.e. the longer bin sizes are ambiguous here). The latter seems most likely, given that there's no change in the peak of the quenching with LC spikes, but it would be helpful to clarify this point.

---

## [Author Response]

Essential revisions:1. The analyses in this study do not directly assess causality or directionality of the interactions reported. This was noted my multiple reviewers, with specific points in the comments below. In addition to these points, in light of the difficulty in claiming causality or directionality from recording data, and it was agreed that there should be a general restructuring of the interpretation to reflect this limitation.

We thank the reviewers for this suggestion. We have revised the text to avoid unsupported suggestions of causality and instead focus on the main measurements and related findings – namely, the relationship between spiking activity in one region relative to spiking activity in the other region. For example, in the second paragraph of the Introduction, we state: “Our aim was to test if and how endogenous, tonic ongoing activity and sensory-driven, evoked, phasic firing patterns responses in the LC relate to changes in neural activity patterns in the anterior cingulate cortex (ACC) of the primate brain.”

a. While the data are very interesting and compelling in their current form. I think that some limitations of the current analyses should be acknowledged. Specifically, the analyses do not allow for inferences about the directionality of effects. So, it remains open if the changes of LC firing rate cause the changes in spike-count correlations in ACC (as seems likely), or vice-versa; or if a third variable causes the effects in both brain regions? Even without causal manipulations, inferences of this kind could be based on an assessment of the temporal relationships of changes in the local signal properties. It could also be based on statistical assessments (e.g. using multivariate autoregressive modeling) of "Granger causality".

We appreciate the reviewer’s comments about causality. We agree that, ideally, explicit causal manipulations (not done in our study) might provide clear answers about the directionality of these effects. In the context of our current data set, we revisited this point and examined temporal relationships between spiking in the two regions. For example, we computed LC spike-triggered average (STA) measures of PETHs of ACC spiking, variability, and pairwise correlations (and ACC STA of LC spiking).

Unfortunately, we were not able to make much sense of these analyses, which are all highly sensitive to even subtle changes in spike rates over the course of fixation that we were not able to compensate for effectively. To overcome this problem, we instead analyzed the time courses of relationships between LC and ACC spiking patterns aligned two fixed task events: fixation onset and beep onset. These analyses, which are presented in the new Figure 9, are consistent with the two main findings from our study: (1) on average, an elevation of baseline LC activity corresponds to a subsequent decrease in ACC r_sc_; and (2) on average, a transient, beep-evoked LC response is followed by a rapid increase in ACC *r_sc_*.

b. Related to "Single-neuron activity during passive fixation"/Figure 2. I'm curious to understand the direction of this effect – does variance in ACC predict spike counts in LC or do spikes in LC predict variance in ACC? Is it possible to look at spike-evoked Fano factor in the ACC (i.e. before and after an LC spike)? Figure 6 is described as implying that these LC spikes and ACC r_sc_ are temporally related, but this is only analyzed as mediated by the relationship between each and pupil size, but this temporal relationship appears not to be investigated directly ("relatively fixed temporal relationships to pupil fluctuations and therefore to each other").

We have updated Figure 2 and the associated text to clarify that we found no statistically reliable relationship between LC spiking and ACC spike count, spike-count variance, or Fano factor. We also analyzed the STAs (with respect to LC spikes) of these quantities and, consistent with what we report in Figure 2, found no reliable results that could allow us to comment on the directionality of these differences (in Fano factor or *r_sc_*).

As noted above, the new Figure 9 more directly compare the time courses of LC and ACC spiking patters, with respect to two fixed task events (fixation onset on no-beep trials and beep onset on beep trials).

c. The authors do nice internal controls testing not only LC-ACC effects, but also ACC-LC effects. They describe that LC-ACC are significant, while the ACC-LC effects are less reliable. This is important for their claims but also as a validation of the analyses. Did the authors formally compare whether there are significant differences between ACC-LC and LC-ACC effects? Showing that ACC-LC is not significant does not address this per se. Slopes (Figure 5) can definitely be tested for LC-ACC vs ACC-LC. (e.g., is the slope more negative in one versus the other).

We thank the reviewers for this suggestion and now include the following new analysis result:

“Moreover, the distributions of these ACC-linked LC *r_sc_* values did not appear to come from the same (shifted) distribution as the LC-linked ACC *r_sc_* values for all five time bins (Kolmogorov–Smirnov test for *H_0_*: both sets of values come from the same distribution, *p* < 0.0189 in all five cases).”

d. Figure 6A, in the legend is described as evidence that LC spikes have a consistent phase relationship with pupil fluctuations that have a period of 600 ms. In the text, this result is taken as evidence that LC peaks 270 ms in advance of the "relevant pupil change" (I'm not entirely clear what pupil phase is being referenced by this phrase). Are these different interpretations? Do LC spikes have a fixed relationship to one component of the pupil fluctuations (like dilation or the cresting at peak size) or are they entrained to the oscillation? Also, it would be good to cite Pong and Fuchs 2000 J Neurophysiol in addition to Joshi, 2016 as evidence for hippus in the monkey at this 1.67 Hz frequency.

We thank the reviewers for raising this point and suggesting the Pong and Fuchs reference about hippus in monkeys.

We have added this reference and also added Lowenstein and Loewenfeld (1969). We have now included a schematic (Figure 6—figure supplement 2) to explain more clearly how these measurements were made. During stable fixation, the pupil size oscillates quasi-periodically. We have also related these findings to those shown in Joshi et al., 2016. In that study (Joshi et al., 2016, Figure 5, LC firing re: pupil events) we showed that LC firing increases relative to maximum rate of pupil dilation and decreases relative to maximum rate of constriction. Here, we corroborated this result (in a new set of experiments and animals/hemispheres) and extended it to show that when aligned to pupil “events” that are not the peak rates of dilation or constriction (i.e., phases other than 0° and 180°), LC spiking show less pronounced peaks. Therefore, during passive viewing, the peak rates of pupil dilation and constriction have a special relationship with LC firing.

e. Related, in Figure 6B, it looks (to my eye) like ACC r_sc_ is also peaking in advance of the trough highlighted in this figure. This would suggest an alternative model, where high r_sc_ in ACC predicts spontaneous LC spiking, and then lower r_sc_ in ACC. This alternative might more in line with Alla Karpova's work that focuses on the effects of ACC on LC activity, rather than the LC-ACC relationship that is the focus of this paper.

We thank the reviewers for this excellent suggestion and agree that the data might support an alternative model, namely that a relative increase in ACC *r_sc_* could predict LC spiking, which in turn could predict reduced ACC *r_sc_*. We have modified the text to reflect this interpretation and added two new figures that provide insight into the relative timing of LC and ACC rsc time-courses (Figure 9).

2. A major caution in interpreting these results is that the paper performs a lot of multiple comparisons in nested bins, and it is not clear that the multiple comparison problem is appropriately controlled for. Many effects appear obvious in the plots suggesting that the three major results would survive correction (i.e. Figure 2G), but for some of the latter analyses it is not clear that effects would survive correction. This problem is complicated by the fact that the tests are strongly interdependent, so it's not clear that a simple correction would be sufficient. It may be more appropriate to conduct permutation tests, or directly ask how independent variables alter how the dependent variable scales with bin sizes.

We thank the reviewers for this helpful suggestion. We have now added an analysis (Figure 4 —figure supplement 1B) to address potential confounds from multiple comparisons. In this shuffled analysis, we calculated the probability of obtaining significant LC-associated changes in ACC *r_s_*_c_ for the real and shuffled data. We found that our results include a far greater set of effects than would be expected by chance, even when doing our interdependent, multiple comparisons.

3. There was some confusion surrounding the motivation for the study as stated in the Introduction. Specifically, the tonic/phasic dichotomy is mentioned in the introduction, but it appears that this was not directly investigated in the rest of the manuscript. Moreover, it was noted that this dichotomy may not be so clear-cut. There may be some overlap with the ideas of ongoing/evoked activity, as in the Aston-Jones and Cohen (2005) framework, however the links between these perspectives are not explored. Reviewers recommend either describing in more detail how the tonic/phasic perspectives motivated the study, or removing this from the introduction and better explaining why one might be curious about the relationship between LC spikes and ACC activity. In the same vein, it would also be helpful to motivate the specific analyses that were performed. Later analyses were well motivated, but the rationale behind the first few were less clear.

We thank the reviewers for this insight. We have revised the text throughout to better motivate the initial analyses and clarify the terminology. We now use the terms “ongoing” and “evoked” instead of “tonic” and “phasic”, as the latter might have distinct implications (as we also discussed in detail in Joshi and Gold, 2020).

4. The authors claim that the firing rate changes in ACC are not reliably related to LC activity, yet the effects appear significant when monkeys are combined? As the authors know, short times scale firing rate correlations (and cross correlations) even in anatomically connected areas are very hard to detect but nonetheless could be real. I think the authors need to take these effects into account to make a convincing case that the other effects they focus on cannot be explained by small population level shifts in rate (or also particularly in fano factor which is highly relevant to population level correlations). I believe this is in their data but it should be fleshed out and the above points should be discussed.

We thank the reviewers for this suggestion. Changes in ACC spiking and Fano factor are indeed *not* reliably related to LC activity (Figure 2). We agree that *any* changes in ACC Fano factor might be important in relation to changes in ACC correlated variability. We explored this via an additional analysis to measure the relationship between LC-linked changes in ACC rsc and ACC Fano factor. This is shown in Figure 4—figure supplement 2. We found a weak but reliable positive correlation between changes in ACC correlated variability and changes in ACC pair Fano factor (Figure 4—figure supplement 2).

5. Two conceptual points were raised that should be addressed in the discussion or elsewhere:a. It is reported that LC phasic activity related to bottom up salient cues (e.g. "surprise" or "startle") increases correlations in ACC, so does this mean the ACC is able to encode less information during surprising events as would be predicted by most theories of correlated activity and information coding in cortex? This has interesting implications for ACC functions, if true.

We thank the reviewers for this suggestion. We have revised and clarified the Discussion related to these results. In particular, increases and decreases in ACC correlations might be associated with different effects. Encoding specific information is one function of cortical networks. Another is coordination of activity, especially when there are sudden, salient changes in the environment (such as a startling event). Reduction in correlations could lead to increased information encoding capacity (as predicted/theorized by many) whereas increased correlations (or greater synchronization) could serve as a coordinating signal within ACC (and possibly between cortical regions, but we have not tested that here), similar to a network reset. A recent report by Valente et al. (2021) suggests a role for increased correlations in behavioral readout.

b. The current literature on LC is very preliminary and theoretic. Precisely, while strong assumptions exist about what it encodes, we don't know if it encodes RPEs, surprises, intense sensory events, higher order RPEs (e.g. some belief state violation related signal), etc. It looks like it is likely complex, however the present task is very simple and may miss important nuances in the LC-ACC network. It is important to point this out and explicitly indicate early in the paper that the procedure is meant to "elicit" LC firing states, rather than test what Lc encodes.

We agree with this important point and have edited the text accordingly. For example, in the Introduction we state that “Our aim was to test if and how endogenous, ongoing activity and sensory-driven, evoked responses in the LC relate to changes in neural activity patterns in the anterior cingulate cortex (ACC) of the primate brain.” In the Discussion, we reiterate that these are measurements made during passive fixation and that similar measurements will need to be made “particularly in the context of goal-directed behaviors” that will test what LC encodes.

6. It was concluded that LC activity has a context-dependent effect on ACC r_sc_, increasing it during passive viewing, but not changing it with LC activity is evoked by a surprising event. However, this conclusion is based on comparing post-stimulus ACC r_sc_ to the pre-stimulus baseline, which does not rule out the alternative interpretation that ACC activity before a surprising event predicts the likelihood of an LC spike. To elaborate, it seems like there is largely a change in the pre-beep, baseline ACC r_sc_ in this data. This would imply that elevated ACC r_sc_ before beep trials predicts no phasic response in LC. If so, the decrease in ACC r_sc_ after LC might be a simple homeostatic effect (i.e. due to the tendency to return to baseline), rather than a context-dependent effect of LC spikes on ACC r_sc_. Did the authors consider this alternative model? Further, the paper does not show that the pattern of ACC r_sc_ after a surprising beep is any different following an LC spike than it is in the absence of an LC spike.

We thank the reviewers for these insightful points. We have re-made the main figure (Figure 7) to show explicitly that the pattern of ACC *r_sc_* after a surprising beep is different following an LC spike than it is in the absence of an LC spike (*’s in Figure 7A). We now also include a new figure (Figure 9) that shows the time course of ACC LC firing rate and ACC *r_sc_* before and after the beep, separated into trials with and without an evoked LC response. We show that: (1) there is no reliable difference in LC spiking or ACC *r_sc_* on trials with versus without a beep-evoked LC response; and (2) the pattern of ACC *r_sc_* after the beep is different on trials with versus without a beep-evoked LC response.

7. In Figure 8, it's not clear if the effects are due to the fact that beep-evoked changes are happening over longer time scales or if they're happening at different latencies relative to the beep (i.e. the longer bin sizes are ambiguous here). The latter seems most likely, given that there's no change in the peak of the quenching with LC spikes, but it would be helpful to clarify this point.

We thank the reviewers for pointing this out. The results shown in Figure 9 address these points by showing the time course of changes in LC spiking and ACC *r_sc_* following the beep.